# Saffron Characterization by a Multidisciplinary Approach

**DOI:** 10.3390/molecules28010042

**Published:** 2022-12-21

**Authors:** Michele Spinelli, Alessandra Biancolillo, Gennaro Battaglia, Martina Foschi, Angela Amoresano, Maria Anna Maggi

**Affiliations:** 1Department of Chemical Sciences, University of Naples Federico II, 80138 Naples, Italy; 2Consorzio Interuniversitario I.N.B.B., Viale Medaglie D’Oro, 00136 Rome, Italy; 3Department of Physical and Chemical Sciences, University of L’Aquila, Via Vetoio, 67100 L’Aquila, Italy; 4Hortus Novus, srl, Via Campo Sportivo, 2, 67050 L’Aquila, Italy

**Keywords:** saffron, crocins, liquid chromatography–mass spectrometry, chemometrics, multiple reaction monitoring

## Abstract

Saffron is a spice obtained from the drying process of the stigmas of the flower Crocus sativus Linnaeus. It is well known that the organoleptic characteristics of this spice are closely linked to the production area and harvesting year. The present work aims to evaluate whether saffron samples produced in different years and origins present sensibly different crocin profiles. To achieve this goal, 120 saffron samples were harvested between 2016 and 2020 in four different Italian areas. The crocins were analysed, identified, and quantified by high-performance liquid chromatography–electrospray–tandem mass spectrometry (HPLC–ESI–MS/MS) in multiple reaction monitoring mode (MRM). Subsequently, ANOVA–simultaneous component analysis (ASCA) was used to evaluate whether the origin and annuity significantly affected the composition of the crocins. ASCA confirmed the relevance of these effects. Eventually, soft independent modelling by class analogy (SIMCA) models were created for each of the four different origins. Mixtures of saffron from different areas were also prepared to test the robustness of the models. SIMCA provided satisfying results; in fact, models provided 100% sensitivity for three origins (Cascia, Sardinia, and Città della Pieve) on the external test set (48 samples) and 88% (sensitivity on the external test set) for the Spoleto class.

## 1. Introduction

Saffron is a spice obtained from the drying process of the stigmas of the flower Crocus sativus Linnaeus. The production process of this spice follows a multistep procedure consisting of several stages: (a) flower harvesting, (b) stigma separation or cleaning, and (c) drying and storage. Each of these steps in the saffron production process is strongly influenced by the traditions of the growing area. The different cultures, together with the climatic characteristics of the area, determine the different chemical compositions that characterize the final product, making it distinguishable from others [1]. However, changes in both preparation procedures and storage methods, and, thus, the year of production, can strongly modify the final composition of the chemical components.

The chemistry of saffron is complex; this spice has primary metabolites that are ubiquitous in nature, such as carbohydrates, minerals, fats, vitamins, amino acids, and proteins. A large number of compounds belonging to different classes of secondary metabolites can be found; they are products of metabolism that are not ubiquitous but are important for the development or reproduction of the organism, such as carotenoids, monoterpenes, and flavonoids, including anthocyanins especially [2]. Carotenoids are the most important constituents of the spice, from which it derives its colour. They include fat-soluble ones, such as α- and β-carotene, lycopene, and zeaxanthin, and water-soluble ones, such as the apocarotenoid crocetin (C_20_H_24_O_4_) and crocins, the polyene esters of the mono- and di-glycoside crocetin. Crocins are an unusually water-soluble family of carotenoids because they are mono- and di-glycosylated esters of the dicarboxylic acid crocetin [3]. They account for 3.5 percent of the weight of the plant’s stigmas. The crocins differ in substituents and configuration (cis and “all trans” crocins), but they are very similar in their physicochemical properties and particularly in polarity. These similarities make their separation and subsequent identification extremely difficult [4,5]. Among the oxidation products of carotenoids, two major compounds can be found: picrocrocin (monoterpene glycoside) and safranal (cyclic monoterpene aldehyde), which are responsible for the bitter and aromatic strength of the spice, respectively. These components are used to define the commercial quality of saffron according to ISO procedures (ISO3632-1, ISO3632-2-2003, International Organization for Standardization) by measuring their absorbance in an aqueous solution at 440 nm (crocin), 257 nm (picrocrocin), and 330 nm (safranal).

Recent studies have reported a distinction between saffron species based on the geographical region [6]. The present work focused on the variances of many types of saffron according to the year of harvest and the place of cultivation within the same country.

By taking advantage of saffron molecular complexity, an attempt was made to obtain a specific target by comparing the crocin composition of the analysed samples. The analyses were conducted on saffron stigmas collected in four different years and from four different geographical areas. This study was based on a high-performance liquid chromatography–electrospray–tandem mass spectrometry (HPLC–ESI–MS/MS) in MRM mode analytical procedure to obtain a rapid, accurate, and reliable analysis. The results obtained allowed us to obtain a separation based on the years of harvesting. Indeed, the content and ratio of the crocins turn out to be such that saffron of different origins can be clearly recognized.

Eventually, the crocin compositions were handled by two different chemometric approaches: ANOVA–simultaneous component analysis (ASCA) [7] and soft independent modelling by class analogy (SIMCA) [8]. ASCA was used to evaluate whether the different harvesting years and origins provided a significant effect on the crocin composition in the samples. On the other hand, SIMCA was exploited to test if it is possible to use the crocin profiles to develop classification models according to the geographical origin of samples. This approach was chosen because it has proven to be a suitable tool for similar aims on the same food matrix [6,9,10,11,12,13].

## 2. Results and Discussion

### 2.1. MRM Analysis

Each sample was analysed by LC–MS/MS in multiple reaction monitoring mode (MRM) in order to characterize the different qualities of the saffron and distinguish them by years of harvesting or origin. The LC–MRM method was based on the selection of peculiar transitions of target molecules by using previous data [14] and optimization of the chromatographic HPLC method by selecting the best eluent and gradient. Table 1 shows MRM instrumental parameters. The sensitivity and selectivity of MRM analysis allowed us to unequivocally determine crocins and crocetin in all saffron samples; in addition, taking advantage of the optimized chromatography gradient, it was possible to separate different isomers (Figure 1) [15].

Relative quantification was performed by using peak areas of the most intense transition (quantifier) and used for statistical analyses.

### 2.2. Chemometric Analysis

The investigated dataset is a complex and multifaceted system (more details are reported in Section 3.1). As described in Section 3.1, the analysed samples were grown in different Italian areas, and some of them were collected over several years. Consequently, the first part of the analysis was restricted to a subset of the available data set in order to assess whether harvesting year had a significant effect on the composition of the crocins given the same geographical origin.

Leaving aside the CP + SP mixed samples, which played a key role in the second part of the chemometric analysis, the samples collected at different time points were those from Spoleto (SP) and Città della Pieve (CP). In fact, the formers were collected in 2016 and 2018 while the CPs were collected in 2016, 2018, and 2020. In order to evaluate the significance of the *year effect*, these samples were subjected to ASCA, and significance was evaluated by permutation tests (10^4^ permutations). The outcome of the analysis is graphically represented in Figure 2. In particular, in the top subplot (Figure 2A), the SC values associated with the Spoleto objects are represented while in the bottom subplot (Figure 2B), the CP samples projected in the space defined by the first two SCs are displayed.

ASCA unveiled that, regardless of the origin, samples significantly differ from one another according to the harvesting year.

Inquiring Figure 2A, a clear division between the 2016 (red bars) and 2018 (green bars) samples is undoubtedly appreciable. In fact, the oldest objects present positive values of SC1 whereas the 2018 samples show the opposite trend.

Similarly, in Figure 2B, is possible to recognize a clear grouping tendency of the samples according to the year. In fact, saffron samples harvested in 2018 (green squares) fall at positive values of SC1 and SC2, in opposition to samples harvested in 2020 which present negative SC1 scores but positive SC2 values. The most peculiar distribution is shown by 2016 individuals (red dots). In fact, part of these objects fall at positive values of the first component while others present slightly negative values of SC1. Nevertheless, both clusters present negative SC2 scores.

The investigation of the loadings of the ASCA models has revealed which compounds contribute the most to the discrimination according to the harvesting year.

Concerning the model associated with the spices harvested in Spoleto, it appeared that seven crocetins drive the model, which are cis crocetin digentiobiose ester, cis crocetin gentiobiosylglucosyl ester, cis 2 and II trans crocetin β D gentobiosyl ester, cis 1 and 2 crocetin β D glucosyl ester, cis crocetin.

On the other hand, the inspection of the loadings associated with the modelling of CP sample, revealed that all compounds except for Crocetin β D glucosyl ester are significant.

After evaluating the *year effect*, the second part of this work focused on estimating whether it is possible to trace samples on the basis of their crocin composition. Consequently, data were autoscaled and four SIMCA models were created, one for each of the investigated origins: Spoleto (SP), Città della Pieve (CP), Cascia (CAS), and Sardegna (SAR).

In order to allow for the external validation of the models, they were divided into a training and test set (by the Duplex algorithm [16]); in particular, the calibration set contained 72 objects (14 from Sardinia, nine from Cascia, 34 from Città della Pieve, and 15 from Spoleto) whereas the validation set included eight samples from Sardinia, three from Cascia, 18 harvested in Città della Pieve, and six from Spoleto. Eventually, in order to furtherly evaluate the robustness of the investigated models, 13 “unknown” samples produced by mixing saffron from Spoleto and Città della Pieve were also included in the validation set for a total of 48 test samples.

Details associated with the calibration models are reported in Table 2 together with the prediction capabilities obtained predicting the test set. The optimal number of PCs was defined in a seven-fold cross-validation procedure; in particular, the complexity which led to the highest efficiency was chosen.

From the table, it is straightforward that the SIMCA models provided excellent results for the Sardinia and Cascia classes and achieved accurate predictions on the other two classes. This can also be observed in Figure 3, where samples are projected onto the $T_{red}^{2}$ and $\;Q_{red}$ space. Inspecting the plots associated with the modelling of the Sardinia class (Figure 3A) and the Cascia class (Figure 3B), the same considerations can be drawn. In fact, in both cases, all the objects associated with the modelled classes (red dots for the Sardinia class and blue squares for the Cascia class) fell inside the threshold (dashed black line) and were accepted whereas all the other samples were properly rejected. Different outcomes can be appreciated in the other two categories, which, apparently, are affected by a greater similarity between one another. Looking at Figure 3C, it is clear almost all CP samples were accepted by the model except for two test objects (over 18) which were erroneously rejected, and, at the same time, one sample belonging to the Spoleto class that was unproperly accepted. What is worth noticing in this model is the fact that all the mixed samples (CP + SP mixtures, represented as black stars in the plots) were accurately rejected. On the other hand, the model of the Spoleto class did not manage to reach the same achievement. In fact, as appreciable from Figure 3D, all the SP samples were properly accepted by the model, but, at the same time, 10 mixtures were also erroneously associated with the Spoleto class. In general, this model appeared not very specific; in fact, besides these mixtures, it also accepted two objects belonging to the CP class. Nevertheless, given the complexity of the classification problem, which was made more arduous by the presence of the mixtures, this can be considered a satisfying result. In general, the possibility that there would be an overlap between Città della Pieve and Spoleto was foreseeable because these two towns are located in two neighbouring areas in the same region (Umbria) with similar pedoclimatic conditions. Moreover, Cascia belongs to Umbria, and it is close to Spoleto, but this town is located at a higher altitude (~650 m.a.s.l) than Città della Pieve (~500 m.a.s.l.) and Spoleto (~400 m.a.s.l.), and this may have created sensible differences in the relative amounts of crocins.

## 3. Materials and Methods

### 3.1. Samples

Saffron samples were provided by Hortus Novus srl and were harvested in different Italian areas. In particular, the samples came from Sardinia (22 samples, coordinates 39°32′59.47″ Nord, 8°47′29.89″ Est), Cascia (Umbria region in Central Italy, coordinates 42°43′3″ Nord, 13°0′56″ Est, 12 samples), Città della Pieve (Umbria region in Central Italy, 42°57′11″ Nord, 12°0′13″ Est, 52 samples), and Spoleto (Umbria region in Central Italy, 42°44′43″ Nord, 12°44′18″ Est, 21 samples). Saffron samples were stored in the dark at room temperature in the absence of humidity.

Samples from Sardinia and Cascia were harvested in 2015. Saffron from Spoleto was collected in 2016 and 2018 whereas samples from Città della Pieve were picked in 2016, 2018, and 2020.

Furthermore, to stress the classification models’ performances and test their efficiency, 13 additional samples were prepared by mixing saffron from Città della Pieve and Spoleto to mimic “unknown” samples.

All solutions and solvents were of the highest purity available and were suitable for LCMS analysis (acetonitrile purchased from VWR Chemicals, methanol and isopropanol purchased from Romil, milli-Q water, formic acid purchased from Fluka).

Sample Preparation

For each sample of saffron (dried stigmas), 10 mg were weighted and suspended in 10 mL of a mixture of 40/40/20 CH_3_OH/ACN/H_2_O *v*/*v*. The extraction was carried out under magnetic stirring for one hour in the dark. The samples were centrifuged at 3000× *g* rcf for 10 min and filtered through 0,45 µm PTFE syringe filters. A total of 1 mL of supernatant was directly transferred into an HPLC autosampler vial, and 1 µL of supernatant was analysed in an LC–MS/MS assay.

### 3.2. LC-MS/MS Analysis

All samples were analysed using a 6420 triple quadrupole system coupled with an HPLC 1100 series binary pump (Agilent, Waldbronn, Germany) with a column Kinetex C18 reverse phase 100 Å 100 × 2.1 mm and 5 μm particle size (Phenomenex) maintained at 20 °C. Mobile phase consisted of solvent A (0.1% formic acid in water) whereas solvent B was 0.1% formic acid in acetonitrile–isopropanol (9:1, *v*/*v*). The linear gradient condition for the HPLC analysis was optimized as follows: 0 to 1 min (10% phase B) in one minute to 30% of B (from 2 to 3 min), then to 50% of B (4 to 5 min), finally to 90% of B for 5 min (6–11), and after that, the column was re-equilibrated for 2 min to the initial conditions; the total run time was 13 min. The flow rate was 200 μL/min. Tandem mass spectrometry was performed by using a turbo ion spray source operated in negative mode, and the MRM mode was used for the selected analytes. Source-dependent parameters of gas temperature, gas flow, nebulizer, and capillary were set at 350 °C, 11 L/min, 40 psi, and 22 nA, respectively. Crocin standard commercially available Crocetin digentobiose ester (purity 99% provided by PhytoLab) was used to validate selected transitions. Data processing was performed by using Agilent MassHunter Quantitative Analysis software (B.05.00). Relative quantification was performed by using peak areas.

### 3.3. ANOVA–Simultaneous Component Analysis (ASCA)

ANOVA–simultaneous component analysis (ASCA) [7] is an explorative tool developed by combining the analysis of variance [17] with simultaneous component analysis (SCA) [18] which allows for investigating the significance of effects on designed data.

Considering a complex system where the significant terms to be taken into account are two (as in the present study), effect α and effect β, the algorithm of ASCA starts with decomposing the original data matrix ***X*** into four matrices, one for each effect ($X_{\alpha }$ and $X_{\beta }$) and their interaction ($X_{\alpha ,\beta }$) and one incorporating the unmodelled variability ($X_{E}$):(1)$$X=X_{\alpha }+X_{\beta \;}+X_{\alpha ,\beta \;}+X_{E\;}$$

Eventually, each matrix in Equation (1) was analysed by SCA. This method can be seen as a particular case of principal component analysis (in particular, it can be depicted as one of its extensions to the multiblock field) and allows for investigating the influence the diverse levels of the DoE explicate on data. Finally, results can be visualized into scores plots by projecting effect by effect and samples onto the space spanned by the first SCs.

### 3.4. Soft Independent Modelling by Class Analogy (SIMCA)

Soft independent modelling by class analogy (SIMCA) [8] is a class-modelling approach developed by Wold and Sjöström in 1977. As with all class-modelling approaches, it allows for individual definition of the class regions of the categories of interest. The approach is based on the assumption that the internal variability in each class can be collected by a principal component analysis (PCA) model [19]. Consequently, the algorithm starts by calculating a PCA model on the training samples belonging to the class of interest. Once this is done, the distance of all the objects from the class model is estimated, and, on the basis of the value they assume, they are accepted (and therefore predicted as appertaining to the modelled class) or rejected. Customarily, a sample is accepted by the model when Equation (2) is verified:(2)$$d=\sqrt{{\left(T_{0.95}^{2}\right)}^{2}+{\left(Q_{0.95}\right)}^{2}}<\sqrt{2}$$
where $T_{0.95}^{2}$ and $Q_{0.95}$ represent the distance in the scores space and the sum of the squared residuals, respectively; as indicated by the subscript, both entities are normalized by the 95th percentile of their distributions [20,21].

If d is greater than the threshold, the sample is rejected by the model and predicted as not belonging to the modelled class.

In general, SIMCA’s results are discussed in terms of sensitivity and specificity. The former represents the percentage of individuals properly accepted by the model while the latter expresses the percentage of the objects correctly rejected. In order to simultaneously maximise both specificity and sensitivity, in the present work, efficiency (i.e., the geometric average of specificity and sensitivity) was used as the definition of the optimal number of PCs to be retained.

## 4. Conclusions

The present work allowed for the investigation of the compositions of some Italian saffron samples produced in different years and locations. ASCA has shown that the year has a significant effect on the crocin composition of saffron grown in the same area. In addition, in the second part of this work, it has been demonstrated the investigated analytes provide suitable information also for tracing these products, indicating that the harvesting area plays a key role in the organoleptic characteristics of this spice. SIMCA models proved to be quite accurate and robust, satisfactorily handling all the individual categories and also the “unknown” CP + SP mixtures.

## Figures and Tables

**Figure 1 molecules-28-00042-f001:**
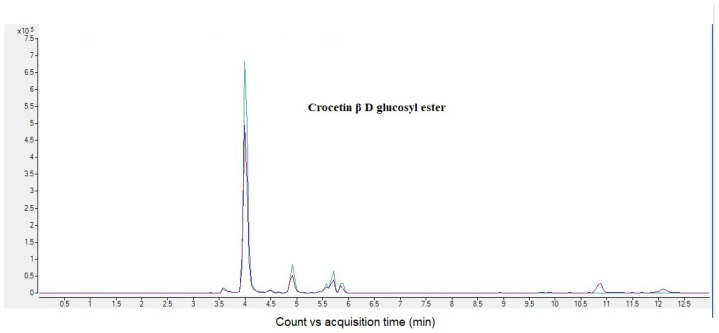
MRM chromatogram of Crocetin β D gentobiosyl ester (Crocin C). It is possible to appreciate different retention times relative to different isomers.

**Figure 2 molecules-28-00042-f002:**
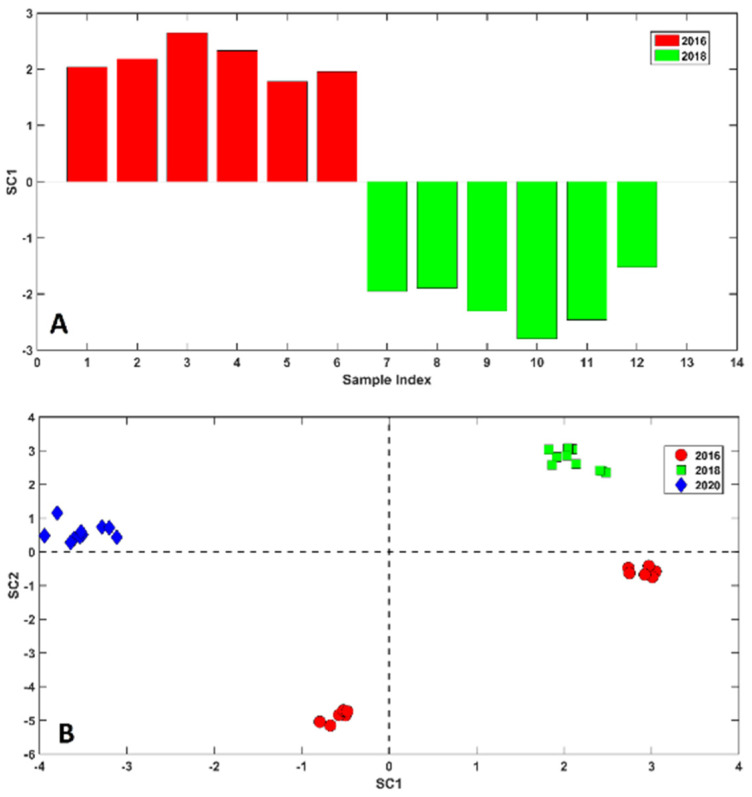
ASCA model of the *year effect*. (**A**) SC values associated with the Spoleto objects. Legend: red bars: saffron harvested in 2016; green bars: saffron harvested in 2018. (**B**) Samples from Città della Pieve projected onto the space defined by the first two SCs. Legend: red dots—2016; green squares—2018; blue diamonds—2020.

**Figure 3 molecules-28-00042-f003:**
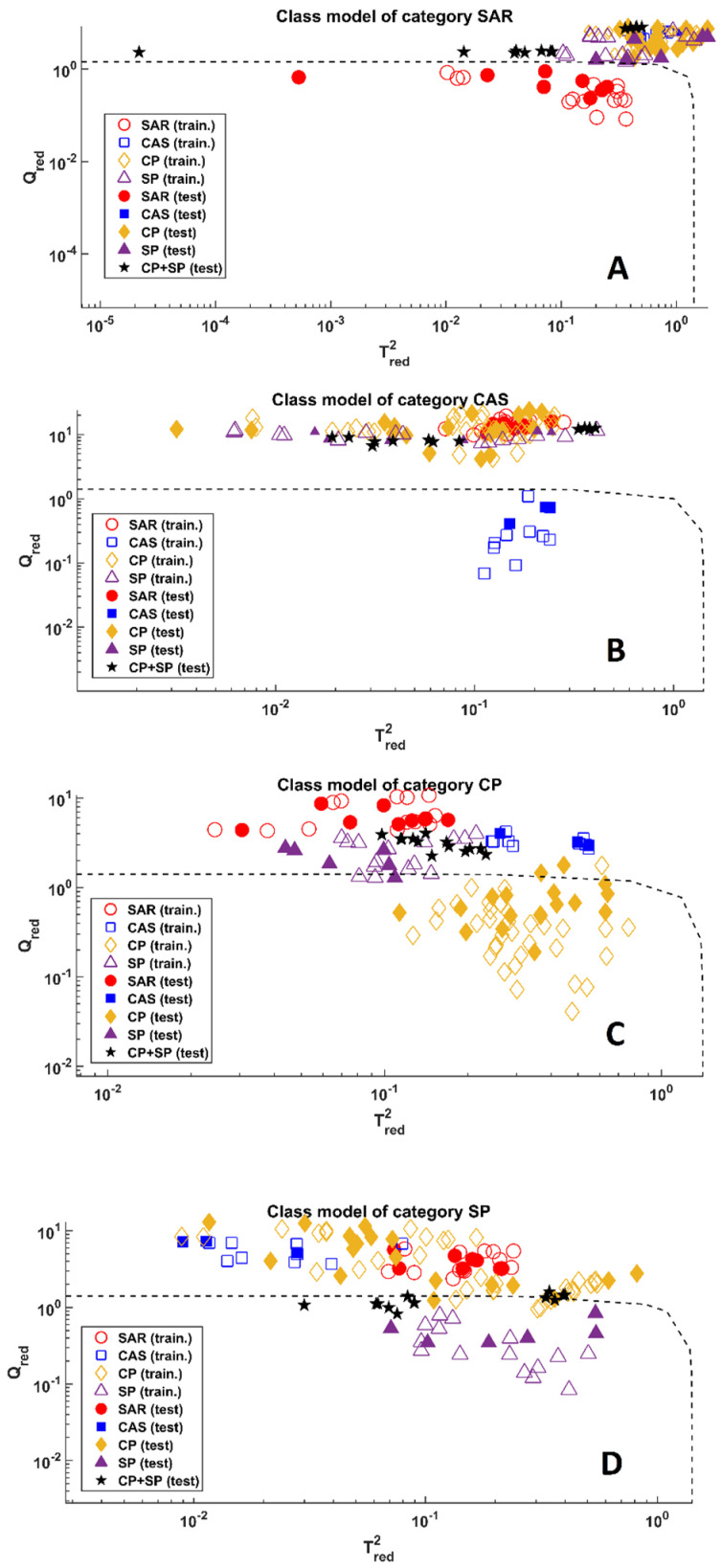
SIMCA analysis, projection of samples onto the $T_{red}^{2}$ and $\;Q_{red}$ space. (**A**) Model of Sardinia Class. (**B**) Model of Cascia Class. (**C**) Model of Città della Pieve Class. (**D**) Model of Spoleto Class. Legend: red dots—Sardinia; blue squares—Cascia; mustard diamonds—Città della Pieve; purple triangles—Spoleto; black stars—mixtures (SP + CP).

**Table 1 molecules-28-00042-t001:** MRM transitions and critical mass spectrometer parameters for target analytes in negative ion mode. (*) Quantifier ion (**) Qualifier ion.

Analytes	Precursor Ion (*m*/*z*)	Product Ion (*m*/*z*)	Collision Energy (eV)
Crocetin	327	283 *	15
		239	20
		165 **	25
Crocetin digentobiose ester	975	652	16
		651 *	20
		327.1 **	20
		59.5	60
Crocetin gentiobiosylglucosyl ester	813	652 *	15
		327.1 **	15
Crocetin β D gentobiosyl ester	651	327.1 *	20
		283	20
		239 **	20
Crocetin β D glucosyl ester	489	327.1 *	15
		324	15
		323 **	15
		283	20
Crocetin gentiobiosyl neapolitanosyl ester	1137	1137 **	5
		813 *	20
Dimethyl_crocetin	355	327.1	15
Picocrocin	329	303 *	15
		285	20
		283	20
		167 **	15

**Table 2 molecules-28-00042-t002:** Results of SIMCA models on the individual classes.

Class Model	PCs	Sensitivity (%CV)	Specificity (%CV)	Efficiency (%CV)	Sensitivity (%Test)	Specificity (%Test)
Sardinia (SAR)	1	100.0	99.2	99.6	100.0	100.0
Cascia (CAS)	1	88.9	100.0	94.3	100.0	100.0
Città della Pieve (CP)	5	91.2	95.5	93.3	88.9	96.7
Spoleto (SP)	2	93.3	94.2	93.8	100.0	71.4

## Data Availability

Not applicable.

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
