# Peer review of "Saffron Characterization by a Multidisciplinary Approach"

_molecules, 2022, doi:10.3390/molecules28010042_

Round 1

Reviewer 1 Report

Manuscript title: Saffron characterization by multidisciplinary approach

Journal: Molecules

Overall comment: Major revision

In this research, the authors have tried with techniques LC-MS/MS, ANOVA-simultaneous component analysis (ASCA) and the help of Soft independent modelling by class analogy, to prove that the ecological location and harvest year (time of storage) of the saffron crop will have an effect on its quality and will lead to changes in its composition of the saffron. Subsequently, samples were collected from four locations in different places in different years and the specified measurements were performed on them.

Although the text of the article was without grammatical errors, there are some concerns in the scientific part of the work that need to be resolved.

From the title of the study, it can be implied that the used techniques were the innovative part of the study. Nevertheless, looking at the Introduction, it seems that the results obtained from the variety of saffron compounds were a novel part. Please revise the introduction to what the main purpose of the work was. If the authors' focus was on using techniques ANOVA-simultaneous component analysis (ASCA) and Soft independent modelling by class analogy to analyze the LC-MS/MS results, in order to determine the variety of saffron compounds, please refer to the works that were used as alternatives methods in the past and the problems of those techniques. It is not necessary to refer to different compounds of saffron in detail.

To evaluate the statistical technique that was used, it would have been better to choose a saffron with a different cultivation period-location (not a saffron obtained from a mixture of two studied saffrons) in order to measure the change in composition.

The process of saffron exploitation, as mentioned in the introduction, is not a complicated process (picking stigma, drying and storage). Please correct this part of the introduction.

In the method section, please specify the geographical coordinates correspond to each location.

According to the authors, the samples were collected in different years and also based on the introduction of the text, the method of keeping saffrons is also very important. But in the methods section, there is no data of how to store the samples. Some of them were collected in 2016, so how did you store them till now? Please provide it.

What is PCA? Full name (principle component analysis) should be given.

Amend this line: “The linear gradient condition for the HPLC 206 analysis gradient was optimized as follows”. Maybe deleting second gradient would make the sentence easier to read.

Author Response

Dear Reviewer
 in the attached document are the answers to your comments.

Best Regards

Reviewer 2 Report

Saffron characterization by multidisciplinary approach

Michele Spinelli 2,3, **Alessandra Biancolillo 1, **, Gennaro Battaglia 2, Martina Foschi1, Angela Amoresano 2,3 and 3 Maria Anna Maggi 1,4*

Report to be sent to the authors:

Dr. Spinelli et al., present a method based on high performance liquid chromatography-electrospray-tandem mass spectrometry for the quantification of crocetins (crocetin, corcetin digentobiose ester, crocetin gentiobiosylglucosyl ester, Crocetin β D gentobiosyl ester, Crocetin β D glucosyl ester, Crocetin gentiobiosyl neapolitanosyl ester, Dimethyl_crocetin, Picocrocin) from saffron stigma collected from different origin and in different years. Then used ANOVA-simultaneous component analysis followed by soft independent modelling by class analogy, to evaluate whether origin and age significantly affect the composition of the corcins.  There is major issue with the experimental design and execution. In its current stage, it is unclear how this method was developed and validated, as well as it is unclear how the concentration of each compound has been quantified without a proper internal standard.

Reviewer’s comments:

Abstract.

1. Line 13, Botanical name for Saffron is Crocus sativus Linnaeus. The first letter of the genus name is capitalized but the specific epithet is not.

2. Lines 17 and 66, Revise “liquid chromatography-mass spectrometry/mass spectrometry” to “high performance liquid chromatography-electrospray-tandem mass spectrometry (HPLC-ESI-MS/MS).

Introduction.

3. Line 65, Storage condition of saffron from 2016, 2018, and 2020 is not provided. Are those samples kept at -80 ËšC freezer? How residual moisture has been removed before storage? Was the drying process the same for all samples? How stable are those compounds of interest in this storage condition?

Results and Discussion

4. Line 84, How was the MRM method optimized?

Table 1.

5. Line 87, Revise Table 1 title. Remove “Results of LC-MS/MS analysis” as this table presents “MRM transitions and critical mass spectrometer parameters for target analytes”.

6. Table 1 title mentioned MRM parameters in “positive ion mode” while last column in the table 1 indicated “negative polarity” for all compounds. Title description should be revised from “positive ion mode” to “negative ion mode” and delete last column “polarity” of the table.

7. A unit for collision energy is missing.

8. Revise “PRECUSOR” to “PRECURSOR”.

9. Product ions for Crocetin digentobiose ester 327,1 and 59,5. Remove comma and report as 327.1 and 59.5. Also, double check the Precursor ion 975 m/z for Crocetin digentobiose ester with molecular weight of 977.

10. Between one to four product ions presented in Table 1 for each analyte. Please clearly mention which one was selected as quantifier ion and which one is the qualifier ion?

11. Which software was used in LC-MS/MS to detect and quantify analytes of interest?

11. Standards information, purity, and manufacturer name are missing in the MRM analysis description. Also, purity of these standards was not stablished. This should have been verified in the laboratory to ensure that the used lot of each standard did not contain impurity. Any purity information provided by the manufacturer?

Figure 1.

12. Line 88, The resolution of the figure is too low.

13. Peaks are not clearly labelled.

14. Identify each compound in the chromatogram. Specify internal standard peak. What is the exact concentration of the compounds in this chromatogram?

15. The size of the character in X and Y-axis is too small to be read. Title of each axis is missing. Revise Figure 1 title description.

16. MS/MS spectra are recommended to be provided for structure confirmation.

Figure 3.

17. Line 176, The resolution of the figure is too low. Symbols are very small.

18. The size of the character in X and Y-axis is too small to be read. Title of each axis is very hard to be read.

19. Graph legends for each graph is missing. If graph legend provided in graph A belongs to all four graphs, please bring it to the top right corner of the graphs sheet outside of all four graphs. 

Sample preparation.

20. Line 196, Please clearly indicate the form of the sample used. Was it dried C. sativus stigmas? Powder form? Or you received partially extracted form from Hortus Novus srl?

21. Line 196-198, At what temperature the extraction procedure was done? What was the basis of choosing 1hr stirring rather than longer stirring time? What was the extraction temperature? It is recommended that the effect of extraction time and temperature on extraction percentage of saffron be investigated.

22. Line 197, Bioactive components of saffron are well known for rapid degradation and isomeric conversion during analysis. Why choosing methanol/water (50:50 v/v) rather than DMSO: methanol (20:80 v/v) for saffron extraction?

23. Line 198, “5000 rpm” should be converted to unit of gravity (X g). For how long each sample was centrifuged?

24. Line 198, Samples were filtered through 0.45 µm filter. Was this filter a Syringeless PTFE filter media with polypropylene housing pour size 0.45 µm? Please provide more information about this filter and manufacturer information.

25. Line 199, What was the volume of supernatant collected after the extraction? Did authors use a uniform volume throughout the extraction- procedure or each sample volume was different? Any internal standard was added to the autosampler before injection?

26. Line 200, Revise “and 1µL of supernatant was analysed in a LC-MS/MS assay” to “and 1µL of supernatant was injected onto the LC-MS/MS”.

LC-MS/MS analysis.

27. Line 202, revise “triple Q” to “triple quadrupole” system coupled with……

28. Line 203-204, revise column information from “column Phenomenex Kinetex C18 reversed-phase 5 μm 100 â„« 100 x 2.1 mm, at the temperature of 20°C” to “column Kinetex C18 reverse phase 100 â„« 100 × 2.1 mm, 5 μm particle size (Phenomenex) maintained at 20°C”.

29. Any guard column attached to the column? Please provide the manufacturer name and size.

30. Line 205, Revise “The eluent A used for the analysis was H2O and 0.1% formic acid, whereas eluent B was a solution of 90% ACN, 10% isopropanol, and 0.1% formic acid” to “Mobile phase, consisting of solvent A (0.1% formic acid in water) whereas solvent B was 0.1% formic acid in acetonitrile–isopropanol (9:1, v/v)”.

31. Line 206, gradient elution described in a confusing way.  It needs rewriting. What was the total run time? Total run time should match with

32. Line 211, revise “negative mode” to “negative ion mode”

Author Response

(The authors gave the same response as above.)

Round 2

Reviewer 1 Report

Accept.